

# Phylogenetic diversity and North Andean block conservation

Omar Daniel Leon-Alvarado[1,2,*] and Daniel R. Miranda-Esquivel[1,*]

[1] Laboratorio de Sistemática y Biogeografía, Escuela de Biología, Facultad de Ciencias, Universidad Industrial de Santander, Bucaramanga, Santander, Colombia
[2] Laboratorio de Sistematica, Entomologia e Biogeografia, Programa de Pós-graduação em Biodiversidade Animal, Universidade Federal de Santa Maria, Santa Maria, Rio Grande do Sul, Brasil
* These authors contributed equally to this work.

Corresponding author
Daniel R. Miranda-Esquivel,
dmiranda@uis.edu.co

## ABSTRACT

**Background:** The Northern Andean Block (NAB) harbors high biodiversity; therefore, it is one of the most important areas in the Neotropics. Nevertheless, the settlement of several human populations has triggered the rapid transformation of ecosystems, leading to the extinction or endangerment of many species.

**Methods:** Because phylogenetic diversity indices quantify the historical distinctness between species, they are adequate tools for evaluating priority conservation areas. We reconstructed 93 phylogenies encompassing 1,252 species and, utilizing their occurrence data sourced from the Global Biodiversity Information Facility, computed the Average Taxonomic Distinctness Index (AvTD) for each grid cell with a spatial resolution of 0.25° within the NAB. The index values for each grid cell were categorized into quantiles, and grid cells displaying values within the upper quantile (Q5) were identified as the most significant in terms of phylogenetic diversity. We also calculated the contribution of endemic species to overall phylogenetic diversity within the NAB, specifically focusing on areas preserved within protected areas.

**Results:** The NAB Andean region exhibited the highest AvTD, with high AvTD values observed in the middle and southern areas of Cordilleras. Endemic species made a relatively modest contribution to the overall phylogenetic diversity of the NAB, accounting for only 1.2% of the total. Despite their relatively small geographical footprints, protected areas within the NAB have emerged as crucial repositories of biodiversity, encompassing 40% of the total phylogenetic diversity in the region.

**Discussion:** Although the NAB Andean region has been identified as the most crucial area in terms of AvTD, some regions in the Amazonian Piedemonte and Pacific lowlands have high AvTD levels. Interestingly, some protected areas have been found to harbor higher AvTDs than expected, given their smaller size. Although the delimitation of new PAs and species richness have been the primary factors driving the expansion of PAs, it is also essential to consider the evolutionary information of species to conserve all aspects of biodiversity, or at least cover most of them. Therefore, using phylogenetic diversity measures and the results of this study can contribute to expanding the PA network and improving the connectivity between PAs. This approach will help conserve different aspects of biodiversity and preserve evolutionary relationships between species.

## INTRODUCTION

The Northern Andean Block (NAB) is a tectonic subplate spanning northeastern Ecuador, Pacific, Caribbean, and Andean regions of Colombia and Venezuelan Andes (*Kellogg et al., 1995*; *Bird, 2003*; Fig. 1). It boasts numerous species, making it one of the most essential areas in the Neotropics (*Myers, 1988*; *Myers et al., 2000*; *Mittermeier, Myers & Mittermeier, 2002*). Its varied climate and resource availability have fostered human settlements over time, turning the NAB into one of the most populated regions in South America (*Goldewijk, 2005*). However, rapid ecosystem transformations have driven the endangerment of many plant and animal species (extinct: 17 species; extinct in the wild: seven species; critically endangered: 496 species; endangered: 1,110 species; data from the IUCN website as of April 14, 2022). Thus, protected areas (PAs) have arisen as an initiative for conservation purposes, with ecological indices used as the primary criteria for their delimitation (*e.g.*, PNN Bahia Portete-Kaurrele; *Diaz-Pulido, 2000*; *Diaz-Pulido & Díaz-Ruíz, 2003*). However, this approach does not consider the history and evolutionary information contained in a phylogeny (*Vane-Wright, Humphries & Williams, 1991*; *Faith, 1992*), implying that critical evolutionary information about species in the area has not been considered in conservation decisions.

From an evolutionary perspective, it is crucial to consider the distinction between species when prioritizing conservation areas. One way to accomplish this is to use phylogenetic diversity indices that evaluate the distinctness between species by quantifying the information contained in a topology (*Vane-Wright, Humphries & Williams, 1991*; *Faith, 1992*; *Schweiger et al., 2008*). These indices can also be used to compare phylogenetic values in different areas (*Vane-Wright, Humphries & Williams, 1991*; *Posadas, Miranda-Esquivel & Crisci, 2001*; *Cué-Bär et al., 2006*).

Phylogenetic diversity indices can help protect not only the most phylogenetically distinct species but also a more significant number of species in fewer areas (*Vane-Wright, Humphries & Williams, 1991*; *Faith, 1992*), mainly when used in conjunction with a complementarity measure (*Cowell & Coddington, 1994*). Despite the potential benefits of using phylogenetic diversity indices for conservation, most studies since 1991 have focused on their applicability and statistical properties (*Vane-Wright, Humphries & Williams, 1991*; *Faith, 1992*; *Redding & Mooers, 2006*), rather than on providing specific recommendations for conservation purposes (*Rolland et al., 2012*; *Winter, Devictor & Schweiger, 2013*). Therefore, developing recommendations for the effective protection of these areas using phylogenetic information is crucial.

Although studies on phylogenetic diversity indices have primarily focused on their applicability and statistical properties since 1991 (*Vane-Wright, Humphries & Williams, 1991*; *Faith, 1992*; *Redding & Mooers, 2006*), there has been limited exploration of their potential application in conservation (*Rolland et al., 2012*). Thus, in this study, our objectives were to (1) present a plausible priority scheme in the NAB for conservation
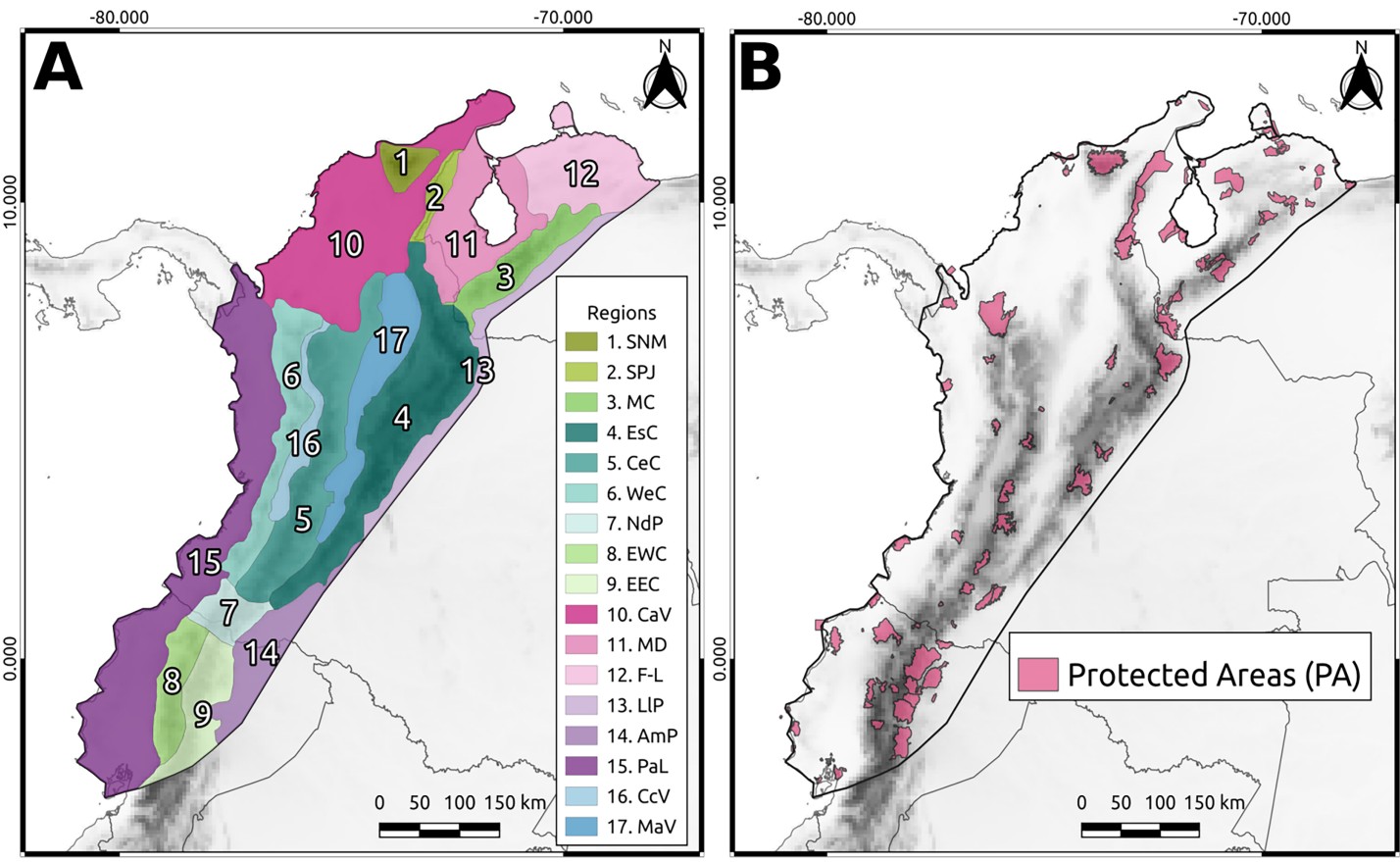

**Figure 1  The North Andean Block (NAB).** (A) Main geographic regions: Sierra Nevada de Santa Marta (SNM), Serranía del Perijá (SPJ), Cordillera de Merida (MC), Eastern Cordillera (EsC), Central Cordillera (CeC), Western Cordillera (WeC), Nudo de los Pastos (NdP), Ecuadorian Western Cordillera (EWC), Ecuadorian Eastern Cordillera (EEC), Caribbean Valley (CaV), Depresión de Maracaibo (MD), Falcón-Lara region (F-L), Llanos Piedemonte (LlP), Amazonia Piedemonte (AmP), Pacific Lowlands (PaL), Cauca Valley (CcV), and Magdalena Valley (MaV). (B) The protected areas from WDPA used in this study.                                                 

purposes, (2) quantify the contribution of endemic species to the phylogenetic diversity of the NAB, and (3) assess the phylogenetic diversity within the PAs.

# MATERIALS AND METHODS

## Taxa and area

The study area was divided into 0.25° grid cells (Fig. 1). To ensure a representative sample of the high biodiversity in the NAB, we selected taxa from different taxonomic ranks (*e.g.*, family, tribe, genus) across different groups (*e.g.*, Vertebrates, Invertebrates, Plants) that had at least three genes available in GenBank for at least 50% of the known species for the taxon. We selected taxa based on the information available in GenBank. We primarily chose families if at least 70% of the species in the family had a minimum of three genes available in GenBank. We used the family as a "taxonomic group"; if not, we moved to the tribe or genus level. The detailed workflow of the taxon selection rules is shown in Fig. S1. To obtain distributional data for the selected taxa (see Table S1), we retrieved their occurrence data (*Telenius, 2011*; see Table S2). Invasive and domestic species were

excluded from the distribution data to prevent bias introduced by human activities, which promotes the widespread distribution of these species.

To assess the reliability of the data distribution used in this study, we calculated the Half-Ignorance Index using the method proposed by *Ruete (2015)*. The index ranges from 0 (no or minimal ignorance) to 1 (maximum or total ignorance). The $O_{0.5}$ parameter (species observation index) was set to a minimum of 10 occurrences to ensure that the presence of a species could be trusted, following the methodology of *Chazdon et al. (1998)*. Using the resulting index values, we generated a raster image that displayed the level of information available for each cell in the study area.

## Phylogenetic reconstruction

For each taxonomic group selected (*e.g.*, family, tribe, and genus; see Table S2), we performed a phylogenetic analysis using the genes and species available for each taxonomic group. Once we obtained the genes from GenBank, we selected the best evolutionary nucleotide model under the Akaike Information Criterion (AIC) (*Akaike, 1974*) for each gene. We performed partitioned reconstruction under the Maximum Likelihood criterion, as implemented in RAxML v8.2.X (*Stamatakis, 2014*), using the nucleotide model GTR +GAMMA for all genes, followed by branch length optimization of the partitioned data in PhyML v3.0 (*Guindon et al., 2010*). For branch length optimization, we used the evolutionary nucleotide model calculated for each gene. Not a single tree in the dataset was dated or calibrated.

## Index selection

We assessed the relationship between the phylogenetic diversity index (PD) and species richness in various areas. To achieve this, we employed two indices: PDi, which relies on a minimum spanning tree approach (*Faith, 1992*), and the average taxonomic distinctness index (AvTD), which is based on pairwise distances (*Warwick & Clarke, 1998*). We calculated PDi and AvTD for each cell in our study area to determine the index that showed the lowest correlation with species richness. Subsequently, we compared the values obtained from each index with the species richness data obtained from these cells. The aim of this analysis was to identify the index with the lowest correlation. For this correlation analysis, we implemented a Bayesian Simple Linear Regression (following *Kruschke, 2014*), as Bayesian analysis outperforms classical analysis (*Wasserstein, Schirm & Lazar, 2019*).

## Prioritization based on phylogenetic diversity

Once an index was selected, the index value of each cell was used to prioritize the grid cells. The index values of the cells were divided and classified into five quantiles, with the highest index values in the upper quantile (Q5) and the lowest in the lower quantile (Q1).

We used a jackknife approach to evaluate the robustness of the phylogenetic data and the resulting prioritization. We randomly selected subsets of the data (*i.e.*, phylogenies) and recalculated the index values. Finally, with the calculated index values, we classified the cells into five quantiles (Q1–Q5). This procedure was repeated 10 times for each subset. The selected subsets were 23 (24.7%), 46 (49.5%), 60 (74.2%), and 93 (100%) phylogenies,

respectively. We used a binomial distribution to estimate the probability of each cell remaining in its initial quantile when all phylogenies were used. This analysis provided a measure of the stability of the prioritization scheme and its sensitivity to variations in the data.

Finally, we measured cell similarity by calculating phylogenetic beta diversity using the method described by *Cardoso et al. (2014)*. Then, using the values of phylogenetic beta diversity, we classified the cells into three different clusters (C1, C2, and C3) based on their similarities using silhouette analysis (*Rousseeuw, 1987*).

### Endemic species

We assessed the endemism of the species by calculating the mean distance between occurrences, and only species with a mean distance of less than 0.25° (~50 km, the distance between the two cells) were considered endemic. Thus, we considered an endemic species as one with a restricted or limited distribution range in the NAB. To understand the impact of endemism on area prioritization, we conducted a jackknife analysis with 100 replicates, randomly removing 25%, 50%, and 75% of endemic species. To assess the influence of endemic species on prioritization, we recalculated the AvTD values for each replicate. Additionally, we evaluated the effect of removing all endemic species from the AvTD.

### Protected areas

To quantify the phylogenetic diversity within PAs (Fig. 1), we identified the grid cells that overlapped with the polygon of each PA to extract and sum their AvTD values. Likewise, we extracted and summed the AvTD values of cells that did not overlap with PAs. To define the polygons of the PAs, we utilized the World Database of Protected Areas (WDPA, *UNEP-WCMC, 2022*).

### Data/script availability

All data and scripts used in this study and more information regarding the methods can be found at https://github.com/oleon12/PhyloDiversity. An interactive map of the results is available at https://rpubs.com/oleon12/PhyloDiv

## RESULTS

### Index selection

We used Bayesian Simple Linear Regression to determine that PDi values correlated highly with species richness, whereas AvTD presented a low correlation value. The mean slope value of the posterior distribution with PDi was 0.78, while AvTD showed a mean slope value of 0.39 (see Fig. S2). Thus, given the low slope value found with AvTD, we selected this index for the prioritization scheme and the other analyses in this study.

### Taxa and data quality

We reconstructed 93 phylogenies at different taxonomic levels, encompassing 1,252 species found in the NAB (see Table S1). The smallest dataset (seven in total) comprised only three genes. In contrast, most taxa were represented by more than four genes, with the Felidae family having the largest dataset of 15 genes (see Table S2). Our results were highly

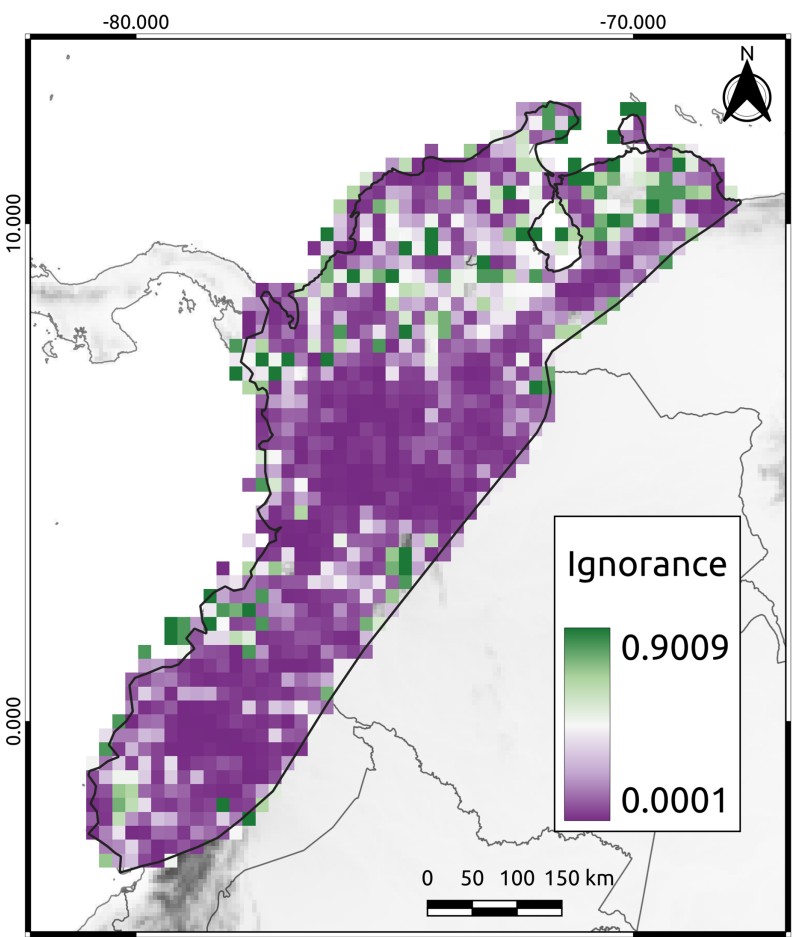

**Figure 2** **The Half-Ignorance index.** The legend depicts the level of ignorance in the NAB. Values closer to one (1) means more ignorance.

robust because adding new phylogenies to the data had little effect on the outcome. With 46 phylogenies (approximately 50% of the total), the mean probability of correct classification was >50%, and with 64 phylogenies, the mean probability was >80% (Table S3).

The mean ignorance value in our analysis was low (mean = 0.26, SD = 0.27), indicating that most cells were well-sampled and had a good representation of the species present (Fig. 2). Only 38% of the cells had high ignition values (red cells in Fig. 2). The AvTD index implemented here considered the presence or absence of species, thus avoiding potential biases that could have arisen from differences in observation efforts. Approximately 44% of the species had 20 or more observations per cell, suggesting that our results were not significantly affected by differences in sampling efforts across the study area.

## Prioritization based on phylogenetic diversity

Q5 cells, which represented the highest priority areas for conservation, accounted for approximately 90% of the phylogenetic diversity of the NAB and covered approximately 20% of the total area. Most of these cells (80%) were in the Andes of Colombia, Ecuador,

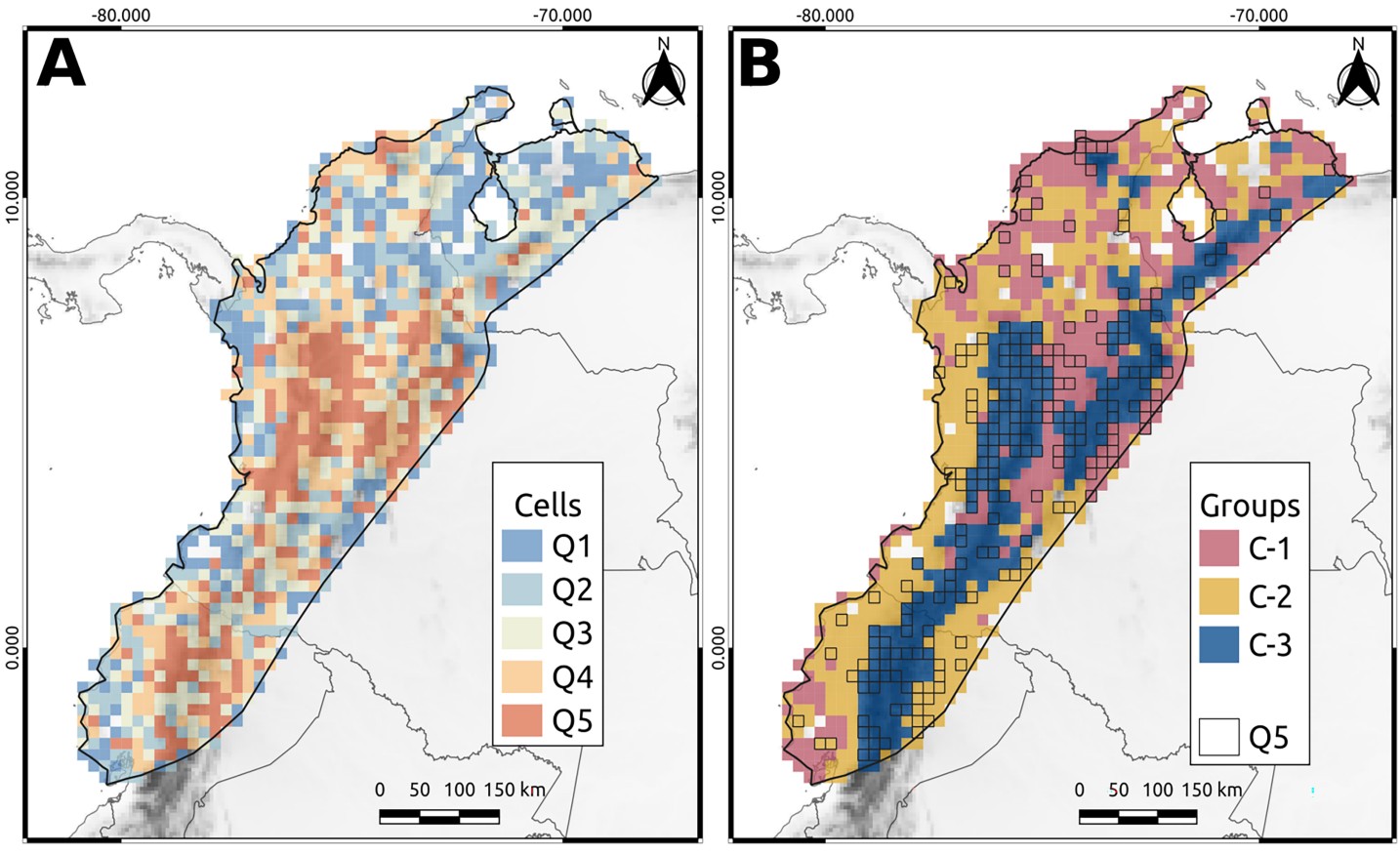

**Figure 3 North andean block phylogenetic diversity.** (A) The cell classification given the average taxonomic distinctness (AvTD) Index. Q5 cells are those with the highest index values. (B) The cluster obtained from the beta-phylogenetic diversity analysis. The lined cells correspond to the Q5 cells.

and Venezuela, whereas the remaining 20% were in the lowlands, particularly the Pacific Ocean (Fig. 3A). In Colombia, Q5 cells were primarily located in the northern and central areas of the western Cordillera (including the Cauca Valley) and eastern Cordillera's western and eastern flanks. The remaining cells were spread in the "Nudo de los Pastos," Caribbean Valley, Sierra Nevada de Santa Marta, Pacific Lowlands, Amazonia, and Llanos Piedemonte (Fig. 3A). Similarly, in Ecuador, Q5 cells were mainly distributed in the Andean region, especially over the western flank of the western Cordillera and the eastern side of the eastern Cordillera. Other Q5 cells were spread over the Pacific and Amazonian Lowlands (Fig. 3A). Finally, Venezuela had a few Q5 cells in the Merida Cordillera, while the others were in the "Depresión del Maracaibo" and Lara-Falcón regions (Fig. 3A).

Silhouette analysis revealed three distinct clusters (C1, C2, and C3) that exhibited specific patterns (Fig. 3B). The first cluster (C1) corresponded principally to the Magdalena Valley, Depresión de Maracaibo, Lara-Falcón region, Llanos Piedemonte, southern Pacific Lowlands in Ecuador, and Caribbean Valley (Fig. 3B). The second cluster (C2) corresponded to the Pacific Lowlands, Amazonian Lowlands, and Piedemonte. In addition, some cells in the second group were distributed in the Caribbean Valley, Depresión de Maracaibo, and Lara-Falcón regions (Fig. 3B). The third group (C3)
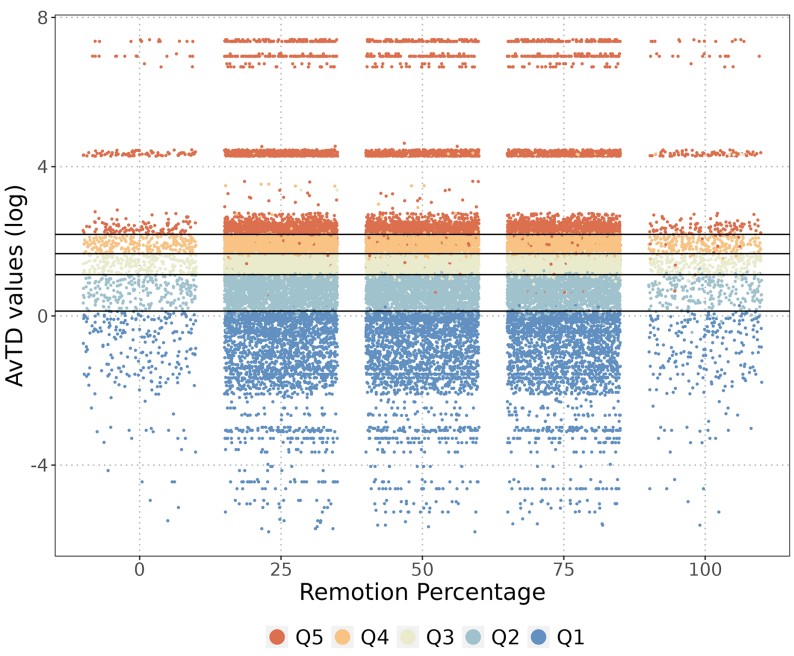

**Figure 4 Endemic species and the phylogenetic diversity.** The cells classification when some percentage of the endemic species were removed. The phylogenetic diversity (AvTD) values of each cell are presented, with colors representing quantile classifications for each removal percentage.

corresponded to the Andean Cordilleras of Colombia, Ecuador, and Venezuela, including Sierra Nevada de Santa Marta and Serranía del Perijá (Fig. 3B).

## Endemic species

We identified 221 species endemic to the NAB. When we completely removed the endemic species from the analysis, the classification of the cells showed some changes, but the original category did not change significantly (Fig. 4). As anticipated, removing endemic species decreased the phylogenetic diversity value of specific cells but did not alter their classification, except for a few cells that changed from Q5–Q2, and *vice versa* (Fig. 4). Removing all 221 endemic species resulted in a marginal decrease of only 1.2% in the overall phylogenetic diversity of the NAB.

## Protected areas

Our analysis of the protected areas in the NAB revealed some interesting patterns. The overall species richness and phylogenetic diversity values were higher outside the protected areas (64% and 67%, respectively). Although we found more species and higher phylogenetic diversity values outside protected areas, they still held a significant amount of the region's total evolutionary information in a relatively small area (9.2% of the NAB's area).

## DISCUSSION

The Andean region harbors the highest phylogenetic diversity in the NAB (Fig. 3A), is considered an essential area for biodiversity (*i.e.*, hotspot; *Myers et al., 2000*; *Mittermeier, Myers & Mittermeier, 2002*), and is a primary target for conservation efforts (*e.g.*, *Ramírez-Villegas et al., 2014*; *Bax & Francesconi, 2019*). Furthermore, *Sechrest et al. (2002)* found more evolutionary information within the Andean hotspot than outside, highlighting its importance for phylogenetic diversity. Although the Andean region stands out as a priority area, our study also revealed other areas outside this region with high AvTD values, such as the Andean valleys and the Amazonian and Pacific lowlands (Fig. 3A). Notably, the Pacific lowlands are within the Tumbes–Chocó–Magdalena hotspot, further emphasizing the importance of these regions for conservation.

Conservation analyses based on a single taxon or specific species groups, such as endemic, small-range, or threatened species, may yield different results. Endemic species are generally characterized by recent diversification, which can result in short branch lengths (*Richardson et al., 2001*; *Davies et al., 2011*). Our study found that endemic species had significantly shorter terminal branch lengths (mean = 0.029, SD = 0.056) than non-endemic species (mean = 0.070, SD = 1.14), and represented only 0.4% of the accumulated branch lengths, resulting in a low contribution to the phylogenetic diversity of the NAB. Although endemic species are more sensitive to environmental disruption and are strong predictors of local extinction risk (*Manne, Brooks & Pimm, 1999*; *Jenkins, Pimm & Joppa, 2013*), they may not accurately predict phylogenetic priorities because of their short branches, which may lead to the underestimation of phylogenetic diversity in a given area. Therefore, it is not advisable to use endemic species as the sole estimator of phylogenetic diversity. It is essential to consider a broader range of taxa and phylogenetic information to accurately identify conservation priorities.

Other phylogenetic diversity indices consider the threat of species extinction, such as the EDGE index (*Isaac et al., 2007*). This index scales phylogenetic diversity based on the IUCN category of the species. We found that approximately 53% of the 1,252 species included in our study were not evaluated by the IUCN (Fig. S3), rendering the EDGE index impractical. Additionally, 55% of the species classified as critically endangered (CR) or endangered (EN) were endemic species with low phylogenetic diversity values and shorter branch lengths. According to the IUCN, high threat levels do not always correlate with high evolutionary distinctiveness (*Caviedes-Solis, Kim & Leaché, 2020*). Therefore, considering that the threat of extinction is vital for conservation, it is essential to consider other factors, such as evolutionary distinctiveness and phylogenetic diversity, to ensure comprehensive and effective conservation planning.

PAs on the NAB have effectively preserved species richness and evolutionary history. However, our study found that species richness and phylogenetic diversity were higher outside PAs than within PAs. This pattern has also been observed in other studies of species richness, phylogenetic diversity, and ecoregions (*Das et al., 2006*; *Zupan et al., 2014*; *Saraiva et al., 2018*; *Castillo et al., 2020*). However, the PAs in the NAB cover a small geographic area (only 9.2% of the NAB area). Although species richness is commonly used

in PA delimitation, evolutionary considerations remain limited (*Sarkar et al., 2006*; *Rolland et al., 2012*; *Lean & Maclaurin, 2016*).

Given that PAs are the primary strategy for conservation (*Virkkala et al., 2019*) and a crucial tool for mitigating the impact of climate change (*Gaüzère, Jiguet & Devictor, 2016*; *Virkkala et al., 2019*), it is necessary to delineate new areas for PAs and improve connectivity between existing ones. Phylogenetic diversity also plays a significant role in this process. Well-connected PA networks can increase species persistence by allowing the movement and flow of different ecological components (*Gaüzère, Jiguet & Devictor, 2016*; *Saura et al., 2017*). Areas with high phylogenetic diversity should be considered when expanding and connecting PAs. For example, in Colombia, Venezuela, and Ecuador, the current PA coverage is approximately 16%, 23%, and 31%, respectively (*Castillo et al., 2020*), making the need for new PAs and the expansion and connection of the current network imperative. Complementarity (*i.e.*, phylogenetic beta diversity) is essential for maximizing phylogenetic diversity. By targeting regions containing cells from all three distinct clusters, as illustrated in Fig. 3, we significantly increased the reservoir of phylogenetic diversity. Notably, this approach does not require all of the selected cells to be within the highest quantile (Q5). The selection of cells originating from diverse clusters is crucial to ensure a substantial infusion of phylogenetic diversity across the spectrum of quantiles ranging from Q1 to Q5. This robust selection mechanism guarantees considerable phylogenetic diversity, thereby minimizing the potential influence of the quantile on the overall outcome.

The current network of protected areas in the NAB needs to adequately represent many ecoregions, including the Magdalena and Cauca Dry Forest, Guajira-Barranquilla Xeric Shrub, and Patia Valley Dry Forest (*Castillo et al., 2020*). Additionally, several ecoregions, such as the Cauca Valley Montane Forest, Eastern Cordillera Real Montane Forest, North Andean Paramo, and Magdalena-Uraba Moist Forest, have high rates of land transformation and degradation and are home to biodiverse and threatened forests (*Bush, Hanselman & Hooghiemstra, 2007*; *Pennington et al., 2010*; *Sánchez-Cuervo et al., 2012*; *Castillo et al., 2020*). Unfortunately, these areas are poorly protected and are underrepresented in the current protected area network. *Echeverría-Londoño et al. (2016)* found that Colombia's Caribbean, Pacific, and Andean regions, which are fully included in the NAB, are the least intact areas with high transformation rates and, as we found, contain remarkable phylogenetic diversity.

Prioritized Q5 cells, which correspond to intact and threatened ecoregions, can help identify critical targets for conservation actions. However, the areas of the NAB that do not correspond to Q5 cells (*i.e.*, Q1–Q4 cells) are not useless, must not be neglected for conservation, and should be the focus of future exploration and checklists. Improving our knowledge of biodiversity in these areas can lead to better conservation decisions in the future. Therefore, a comprehensive strategy for conservation in the NAB should include expanding and connecting the current network of PAs to better represent poorly protected ecoregions and to prioritize conservation efforts in intact and threatened ecoregions.

## CONCLUSIONS

The Andean region of the NAB may have the highest AvTD values; however, it is crucial to recognize significant phylogenetic diversity in the Amazonian Piedemonte and Pacific lowlands. Conservation efforts should be comprehensive and consider all aspects of biodiversity. Therefore, even with the low phylogenetic diversity of endemic species, they must be used with caution for conservation purposes. Species richness was the primary index used for PA delimitation. Given the small geographical area of PAs, they protect a substantial amount of phylogenetic diversity. However, indices other than richness should be considered to effectively expand and delimit PAs. Although there is debate on the importance of evolutionary aspects in PA delimitation and conservation planning, preserving the evolutionary potential of the biota is crucial (*Vane-Wright, Humphries & Williams, 1991*; *Faith, 1992*; *Forest et al., 2007*; *Saraiva et al., 2018*). This study aimed to provide valuable information to experts, non-experts, and governments, encouraging the increased use of phylogenetic diversity in conservation actions and emphasizing the significance of the evolutionary framework in conservation decisions.

## ACKNOWLEDGEMENTS

We thank Karen Mendez and Juan David Bayona for their valuable comments on the manuscript. We also thank all members of the Laboratorio de Sistemática y Biogeografía, Universidad Industrial de Santander, for their continuous feedback on this study.

### Funding

Daniel R Miranda-Esquivel received funding support from the following sources to purchase computational equipment, maintenance, and furniture: 'Una expedición para reducir el déficit de conocimiento en biodiversidad a una escala regional en Santander, Colombia.' Código: 1102-866-75870 fondo nacional de financiamiento para la ciencia, la tecnología y la innovación Francisco José de Caldas, Ministerio de Ciencia, Tecnología e Innovación de Colombia – Colombia Bio (VIE-UIS 8034), and 'Inventario de la diversidad biológica en una región del sur de Bolívar, Colombia' (VIEUIS 8867), Ministerio de Ciencia, Tecnología e Innovación, Ministerio de Educación Nacional, Ministerio de Industria, Comercio y Turismo, and ICETEX, Programa Ecosistema Científico-Colombia Científica from Fondo Francisco José de Caldas, Grant RC-FP44842212-2018. The funders had no role in study design, data collection and analysis, decision to publish, or preparation of the manuscript.

### Competing Interests

The authors declare that they have no competing interests.
## Author Contributions

- Omar Daniel Leon-Alvarado conceived and designed the experiments, performed the experiments, analyzed the data, prepared figures and/or tables, authored or reviewed drafts of the article, and approved the final draft.
- Daniel R Miranda-Esquivel conceived and designed the experiments, performed the experiments, analyzed the data, prepared figures and/or tables, authored or reviewed drafts of the article, and approved the final draft.

## Data Availability

All data, and scripts used in this work are available at GitHub and Zenodo:

- https://github.com/oleon12/PhyloDiversity

- Omar Daniel, L.-A., & Daniel Rafael, M.-E. (2023). Phylogenetic diversity and North Andean block conservation - Data [Data set]. Zenodo. https://doi.org/10.5281/zenodo.10140032

## Supplemental Information

Supplemental information for this article can be found online at http://dx.doi.org/10.7717/peerj.16565#supplemental-information.

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
