# Peer review of "Phylogenetic diversity and North Andean block conservation"

_PeerJ, doi:10.7717/peerj.16565_

## Round 0.1 · original submission · Major Revisions

The manuscript has received three very thorough reviews. Two reviewers point to serious flaws in the study analysis, in particular not taking alpha diversity of species into account when calculating phylogenetic diversity. Additional improvements involve including a full species list and a detailed map showing the study area. These and other issues will need to be addressed for the manuscript to be considered for re-review.

·

Basic reporting

Dear Editor and authors,

I have now read and reviewed the ms entitled “Phylogenetic diversity and the North Andean Block conservation” which has been submitted to PeerJ.

This manuscript deals with the species phylogenetic diversity distribution in the Northern Andean Block and how the protected areas delimited in the region act to protect this diversity. Overall, this manuscript potentially presents important and very welcoming data on the phylogenetic diversity in this portion of the Andes and could be used as baseline data for future conservation plans to the area.

The manuscript presents a clear language, and professional English was used throughout the text, minor grammatical errors were found and pointed out during the review. The structure presented throughout the manuscript follows the pattern recommended by PeerJ.
In its current form, however, I feel it needs a rearrangement in the text, and an improvement for clarity in the methods, since this directly implies in the understanding of the results presented. I would strongly recommend that the authors make a more detailed inspection in the organization of the paragraphs in the introduction, since more than one paragraph appears with repeated information throughout the text. There is also, a lack of important figures in the work, such as a map of the study region and the preserved areas used in the analyses, etc. figures that do not necessarily need to be in the main text, but that need to be presented at least in the supplementary material for a better understanding of the work.

Having made the above suggestions, I would include below some limitations of their current approach that I believe need to be improved, or at least taken into account, in future versions of this manuscript.

Specific comments:

50. A map elucidating the study region and its location in the Andean context would be very interesting.
57. This sentence needs more detail. It is not clear what this information refers to? animal, plant species?
75. This sentence needs more detail, what is considered a distinctive species in here? I suggest a better explanation of the term.
75-76 / 77-82. You are presenting repeated information in the last three paragraphs of the Introduction. I suggest removing the third paragraph and keeping the current fourth and fifth.
104. The authors present a “Taxa and area” section in the text, but neither in the main text nor in the supplementary materials is it possible to identify which species were selected, a list of species or a better explanation beyond the presented figure S1 would be of good use.
127-147. It is not clear what is considered a taxonomic group here? the genres used? were three genes selected for each genus? Were phylogenies constructed from these data? presenting the phylogenies used at least in the supplementary material would be important for a better understanding of the data.
135. Your Methods need more detail. I suggest that you improve the description of what statistical analyzes were used to obtain the results of the study, as it is not explained in the text or in the supplementary material and only available by checking the GitHub.
158. It was not clear the classification of endemic species here. Are endemic species, only species with limited range?
182. Was a phylogeny built for each genus used? Did you use the whole genus to construct the phylogenies trees? or just the species occurring inside the NAB? is not easy to understand just reading the main text and accessing the supplementary material.
193. There is a typo in here, avoid
253. The authors state that the endemism may lead to underestimation of phylogenetic diversity in a given area and should not be usable as a sole estimator of phylogenetic diversity in a given area. I was surprised to not find a detailed list of species used in the manuscript, or at least the final number of the species used, including species who were considered endemic in this study, even if it comes as an appendix or supplementary material.

Experimental design

The authors present here original primary research who is totally in the scope of the journal, the motivation and the question of the manuscript is valid and presents itself as novelty to the study area. However, I think that the methods are not described with sufficient detail, and some important information necessary to understand the data used are lacking and in need of being supplemented. I also recommend a better description of the analyses in the main text; it is not well described which kind of analyses were used to obtain the results of the work, and it is only possible to understand what was done and how the results were obtained by checking the GitHub link provided by the authors.

Validity of the findings

The manuscript offers the results of the phylogenetic diversity values across the Northern Andean Block, with some of the protected areas analyzed harboring a high amount of this diversity. The idea that permeates the text is that the work could be used in the future as a tool in conservation decisions elucidating the role of phylogenetic diversity besides species richness to protected areas delimitation. The conclusions presented in the main text are linked to the original research question and I totally agree with the authors that this work could be used as baseline information in future conservation plans. I really hope the authors manage to get this very important work published.

Additional comments

I would like to congratulate the authors for the extremely interesting and important work present in here, and I hope to see it published soon.

Reviewer 2 ·

Basic reporting

Dear authors, I have reviewed the manuscript by Leon-Alvarado and Miranda-Esquivel entitled “Phylogenetic Diversity and the North Andean Block Conservation” submitted for consideration at PeerJ. The English language should be improved to ensure that an international audience can clearly understand your text. A few examples where the language could be improved include lines 54 and 57 – the current phrasing makes comprehension difficult. The literature included could be improved as well, such as see Tucker et al. (2017) Biological Reviews “A guide to phylogenetic metrics for conservation, community ecology and macroecology”. Figures should have an inset showing the study area on a map of the entire Neotropics.

Experimental design

Phylogenetic diversity is highly correlated with species richness, thus if this effect is not considered, results can reflect the effect of species richness rather than the effect of phylogenetic history in generating geographic patterns. Several papers have reported this issue and there are methods available to estimate standardize effect sample size of phylogenetic diversity for a given area, which would remove the effect of species richness. Please clarify why the authors chose to assemble data from GenBank and estimate a phylogeny when they could have used more robust fully sampled phylogenies, that are time-calibrated and available in the literature.

Validity of the findings

Given that the methods used could be improved, the findings of the present papers will have to be re-evaluated.

Reviewer 3 ·

Basic reporting

I’ve read the manuscript ID PeerJ 85796v1 entitled “Phylogenetic diversity and the North Andean Block conservation.” This manuscript represents an attempt to characterize the phylogenetic diversity (PD) of the NAB region and to identify areas for conservation based on the PD outcomes. Notably, the authors evaluated several taxonomic groups using species distributions from GBIF and molecular phylogenies constructed using data from GenBank. Overall, the results are interesting—albeit not novel—and the methods seem to be implemented properly; however, I found myself wanting to see more of the data to understand what might be revealed by these analyses fully. For example, I find it a bit difficult to articulate some of the results given the data completeness—i.e., 38% of the cells present high ignorance.

Experimental design

The authors applied several approaches to ensure the validity of their results, including the Half-Ignorance Index. They also reconstructed the phylogenetic relationships of several taxa. However, very few details are provided regarding the data completeness. The authors present a summary table of the data in the supplementary material, but it is difficult to know what information they are providing without a table description.

Also, it is not specified if the phylogenetic trees were dated or not. This information must be clear in order to avoid confusion among the readership.

My main concern about the methodology is the selection of PD quantiles as a prioritization scheme. Again, very few details are provided about this process. Specifically, what does mean low and high PD values in the context of spatial prioritization? Some cells may present low PD (Q1), but that does not mean that those cells are poor in terms of number of species. Another scenario would be that high PD (Q5) is caused by few species that present long branches. This is not a small issue and must be clarified in the methodology or introduction section.

I understand the importance of including a protected areas (PA) analysis; however, it is unclear how the analyses were performed. Did the authors calculate PD considering only species within each PD? Or they just extracted the PD values that fall within the PA polygons?

Validity of the findings

The findings are interesting, but it is difficult to endorse the authors’ findings without tackling the points provided above.

Also, considering that this paper is trying to reach conservation practitioners, managers, and policymakers, the way that the results are presented is a bit dangerous. Focusing conservation efforts on high PD cells is quite circular to previous findings and also would stimulate the neglection of cells with low PD. I strongly recommend the authors to reframe or expand the description, interpretations, and implications of their results so that they are not misinterpreted.

---

## Round 0.2 · Minor Revisions

I have received two reviews of your revised manuscript. Both reviewers are favorable, although reviewer 2 does have some significant comments. In particular, reviewer 2 advocates using phylogenetic methods in your study. I realize that this could be a a major effort on your part. I would suggest trying to implement at least one phylogenetic method as reviewer 2 suggests. Alternatively, if you cannot do that, please devote a paragraph in your discussion describing how phylogenetics methods could have improved the study and overcome some common statistical problems. This will make it clear that the phylogenetic approaches are preferred, even if you didn't implement them.

·

Basic reporting

Dear Editor and authors,

I have now read the revised version of the manuscript entitled “Phylogenetic diversity and the North Andean Block conservation” which has been resubmitted to PeerJ, and I am aware of all changes that have been made to the manuscript.
I am happy with all the changes made by the authors and the attention they paid to all the aspects mentioned by the reviewers. The manuscript now presents significant improvements not only in the text, which is now clearer to the reader, but also in the methods section, which now presents more information, and in the new results, which were modified taking into account reviewers' suggestions.
The addition of new figures, clarity in the methods used and greater robustness in the supplementary materials presented certainly make this manuscript a great contribution to knowledge and conservation in the Northern Andean Block.

Experimental design

no comment

Validity of the findings

not comment

Additional comments

not comment

Reviewer 2 ·

Basic reporting

This revised manuscript marks a significant enhancement over the earlier version that I had the privilege to review. The textual improvements are readily apparent. Nevertheless, I would like to offer some constructive suggestions to further align the manuscript with its stated objectives. I am confident that the authors will find this feedback valuable as they continue to refine their work for eventual publication.

Experimental design

First and foremost, I recommend employing a phylogenetic index that minimizes potential biases associated with diversity summary metrics, such as the strong influence of species richness. The manuscript aims to investigate the distinctness between species within a phylogenetic context. To achieve this, I suggest utilizing the Mean Phylogenetic Distance (MPD) along with its standard deviation for each grid cell, as opposed to non-phylogenetic metrics like the Average Taxonomic Distinctness Index (AvTD). This adjustment aligns the analyses more closely with the manuscript's primary objective of quantifying distinctness through phylogenetic means.

Furthermore, I recommend refining the calculation of endemicity. This could be achieved by assessing the frequency of species across grid cells, employing defined thresholds. For example, the manuscript could establish a predetermined threshold based on the number of grid cells occupied by a species to classify it as endemic. This threshold might range from 1 to 5 grid cells or could be determined by estimating quantiles using data from the 1252 species under study, with a focus on selecting the lower quantile to identify endemic species. Such refinement in the definition of endemicity would contribute to the manuscript's precision and scientific robustness.

Validity of the findings

It is crucial to underscore that AvTD is not a phylogenetic method. Consequently, any figures and interpretations that rely on AvTD should be appropriately adjusted to ensure that they align with the phylogenetic focus of the study.

Annotated reviews are not available for download in order to protect the identity of reviewers who chose to remain anonymous.

---

## Round 0.3 · accepted · Accept

Thank you for incorporating this last round of revisions. I am happy to recommend publication for your paper at this stage.